# Improvement of Retinal Microcirculation after Pulmonary Vein Isolation in Patients with Atrial Fibrillation—An Optical Coherence Tomography Angiography Study

**DOI:** 10.3390/diagnostics12010038

**Published:** 2021-12-24

**Authors:** Philipp S. Lange, Natasa Mihailovic, Eliane Esser, Gerrit Frommeyer, Alicia J. Fischer, Niklas Bode, Dennis Höwel, Friederike Rosenberger, Nicole Eter, Lars Eckardt, Larissa Lahme, Maged Alnawaiseh

**Affiliations:** 1Division of Electrophysiology, Department of Cardiovascular Medicine, University Hospital Muenster, D-48149 Muenster, Germany; gerrit.frommeyer@ukmuenster.de (G.F.); alicia.fischer@ukmuenster.de (A.J.F.); niklas.bode@ukmuenster.de (N.B.); dennis.hoewel@krupp-krankenhaus.de (D.H.); lars.eckardt@ukmuenster.de (L.E.); 2Department of Ophthalmology, Hospital Fulda, University of Marburg, Campus Fulda, D-36043 Fulda, Germany; nat.mihailovic@gmail.com (N.M.); maged.alnawaiseh@klinikum-fulda.de (M.A.); 3Department of Ophthalmology, University of Muenster Medical Center, D-48149 Muenster, Germany; eliane.esser@ukmuenster.de (E.E.); friederike.rosenberger@gmx.de (F.R.); nicole.eter@ukmuenster.de (N.E.); larissa.lahme@ukmuenster.de (L.L.)

**Keywords:** atrial fibrillation, OCTA, optical coherence tomography angiography, optic nerve head perfusion, retinal perfusion

## Abstract

Purpose: To evaluate retinal and optic nerve head (ONH) perfusion in patients with atrial fibrillation (AF) before and after catheter ablation of AF with pulmonary vein isolation (PVI). Methods: 34 eyes of 34 patients with AF and 35 eyes of 35 healthy subjects were included in this study. Flow density data were obtained using spectral-domain OCT-A (RTVue XR Avanti with AngioVue, Optovue, Inc, Fremont, California, USA). The data of the superficial and deep vascular layers of the macula and the ONH (radial peripapillary capillary network, RPC) before and after PVI were extracted and analysed. Results: The flow density in the superficial OCT-angiogram (whole en face) and the ONH (RPC) in patients with AF was significantly lower compared to healthy controls (OCT-A superficial: study group: 48.77 (45.19; 52.12)%; control group: 53.01 (50.00; 54.25)%; *p* < 0.001; ONH: study group: 51.82 (48.41; 54.03)%; control group: 56.00 (54.35; 57.70)%; *p* < 0.001;). The flow density in the ONH (RPC) improved significantly in the study group following PVI (before: 51.82 (48.41; 54.03)%; after: 52.49 (50.34; 55.62)%; *p* = 0.007). Conclusions: Patients with AF showed altered ocular perfusion as measured using OCTA when compared with healthy controls. Rhythm control using PVI significantly improved ocular perfusion as measured using OCT-A. Non-contact imaging using OCTA provides novel information about the central global microperfusion of patients with AF.

## 1. Introduction

Atrial fibrillation (AF) is the most common arrhythmia in industrialized countries with an increasing burden of morbidity. Pulmonary vein isolation (PVI) has evolved as an effective therapeutic option for patients with symptomatic AF [1]. Today, PVI is a cornerstone of modern AF management [2] showing success rates of up to 70%. However, PVI is associated with a potential risk of cerebral ischemic events and a low but measurable risk of acute stroke [3]. In addition to clinically apparent neurological deficits, Doppler studies during PVI procedures have shown asymptomatic micro-embolization events during ablation [4]. Correspondingly, imaging studies using cerebral MRI have demonstrated a high rate of newly occurring asymptomatic cerebral lesions following PVI [5].

Cerebral perfusion seems to be altered in patients with AF. AF is assumed to play a role as a possible contributor to cerebral hypoperfusion and cognitive dysfunction [6]. In fact, convincing evidence suggests that cerebral perfusion declines with worsening cardiac function, and impaired cardiac hemodynamics caused by AF may further suppress cardiac output and cerebral blood flow [7]. Recently, Gardarsdottir et al. demonstrated a decreased cerebral blood flow and brain perfusion in AF patients compared to those with sinus rhythm using MRI imaging [8].

Optical coherence tomography angiography (OCT-A) is a novel imaging technology which provides high-quality images of the retinal vascular and vasculature of the optic nerve head (ONH) [9,10,11,12]. OCT-A is non-invasive, fast, and reproducible imaging modality. In a previous study by our group, we found a reduced retinal and ONH perfusion in patients with AF compared to healthy controls [13].

The aim of this study was to evaluate changes in ocular perfusion in patients with AF using OCT angiography imaging before and after rhythm control therapy using PVI.

## 2. Methods

### 2.1. Study Population

In this study, 34 patients over 18 years with symptomatic atrial fibrillation and planned for pulmonary vein isolation PVI at the Department of Cardiology II—Electrophysiology University of Muenster Medical Center were consecutively enrolled (“study group”). The diagnosis of AF was made by expert cardiologists and based on documented ECG recordings. A group of 35 healthy patients without any history of heart disease or any other condition served as a control group (“control group”).

Patients with media opacities preventing high-quality imaging, (vitreo-)retinal pathologies or neurological diseases were not considered as study participants. OCT-A imaging was performed before and one day after pulmonary vein isolation.

All patients in the study group received transesophageal echocardiography in order to rule out intracardiac thrombus formation immediately before pulmonary vein isolation. As described elsewhere in detail [14,15,16], pulmonary vein isolation was performed using a cryoballon or by radiofrequency ablation, respectively. A steerable decapolar catheter (Lifewire, St. Jude Medical, North Chicago, Illinois, USA) was placed in the coronary sinus. Transseptal puncture (TSP) was performed under fluoroscopic imaging. Heparin was administered with a target active clotting time (ACT) of >300 s. The pulmonary veins were visualized by injecting a nonionic contrast through a multipurpose catheter. If patients were in AF, sinus rhythm was restored by electrical cardioversion. In case of the cryoballoon PVI, occlusion of the PV was tested by contrasting agent injection. Freezes were delivered in each vein, and the isolation of the right PVs was performed under continuous phrenic nerve stimulation. If PVI was performed using RF energy, a 3D map of the left atrium and the pulmonary veins was generated followed by isolation of the pulmonary veins by antral circumferential RF ablation.

### 2.2. Anticoagulation

Before PVI, patients in the study group received oral anticoagulation according to their individual thromboembolic risk assessed by the CHA_2_DS_2_-VASc score in accordance with current guidelines [1]. Due to a relevant CHA_2_DS_2_-VASc score, 26 patients of the study group received oral anticoagulation before PVI. Following PVI, all patients received an oral anticoagulation.

### 2.3. Examination

Before PVI, patients underwent a complete ophthalmic examination including refraction, visual acuity, and intraocular pressure (IOP) measurements, slit lamp biomicroscopy, retinoscopy, fundus photographs, and OCT-A imaging.

### 2.4. OCT-A

Using decorrelation between consecutive structure OCT scans, it is possible to visualize retinal vasculature without intravenously injected dye. The OCT-A technology has been described in detail elsewhere [9,17]. Briefly, repeated OCT scans of a certain area are performed, and the OCT images of that area are evaluated to identify possible changes. Blood flow in the retinal vessels will result in changes between the successive OCT images, whereas static tissue will show no change. OCT-A imaging in this study was performed using the AngioVue device (RTVue XR Avanti with AngioVue, Optovue Inc, Fremont, CA, USA). This system uses the split-spectrum amplitude-decorrelation angiography (SSADA) algorithm to generate the angiography data with an A-scan rate of 70,000 scans per second.

### 2.5. Scans

Macula imaging was performed with an 8 × 8 mm^2^ scan and a 3.0 × 3.0 mm^2^ scan. Imaging of the optic nerve head (ONH) was performed with a 4.5 × 4.5 mm^2^ scan. The 3.0 × 3.0 mm^2^ scan of the macula and the 4.5 × 4.5 mm^2^ scan of the ONH were used for quantitative analysis. The 8 × 8 mm^2^ scan was used to visualize retinal vasculature. OCT-A imaging was performed by an expert examiner under the same conditions in the same location. One patient with poor quality images (related to poor signal strength or relevant motion artifacts such as white lines or roiling distortion) was excluded from the quantitative analysis. Both ONH and macular OCT-angiograms were automatically segmented into 4 tissue layers by the included software. The ONH angiogram was segmented into optic nerve head, vitreous, radial peripapillary capillary (RPC), and choroid, and the macular OCT angiogram was segmented into superficial, deep, outer retina, and choriocapillaris. The segmentations were checked for accuracy by an expert (NM, MA, LL, EE) and the flow density data of the radial peripapillary capillary (RPC) and the superficial and deep vascular layer of the macula were then extracted and analyzed. Retinoscopy, fundus photography, and OCT-A images were evaluated by an expert examiner (EE, LL, MA, NM) in order to detect microinfarction.

### 2.6. Data Analysis and Statistics

IBM SPSS^®^ Statistics 27 for Windows (IBM Corporation, Somers, NY, USA) was used for statistical analyses. The data were tested for normality distribution using the Shapiro–Wilk test and the flow density data was found to not fit a normal distribution. The differences between the AF group and the control group were assessed using the independent Student’s t-tests for normally distributed variables and the Mann–Whitney U test for non-normally distributed variables. Changes in the AF group after PVI compared with the baseline were assessed using the Wilcoxon signed-rank test. The data are presented as median (25, 75 percentile deviation). All inferential statistics are intended to be exploratory, not confirmatory, and are interpreted accordingly. The global statistical significance level was set to 0.05.

## 3. Results

We initiated our study with a comparison between the study group (before PVI) and the healthy control group. There was no significant difference in age between the study group (AF) and the control group (study group: 60.50 (52.00; 65.25) years; control group: 50.70 (51.00; 70.00) years; *p* = 0.908). The clinical characteristics of the study group and the control group are summarized in Table 1.

The flow density (whole *en face*) in the ONH (RPC) in the study group was significantly lower compared to the control group (study group: 51.82 (48.41; 54.03)%; control group: 56.00 (54.35; 57.70)%; *p* < 0.001). Flow density data in the RPC of the ONH, and the superficial/deep retinal OCT angiogram of the macula are summarized in Table 2.

Next, we analyzed the impact of AF ablation by PVI on retinal perfusion in the patients of the study group. The flow density (whole *en face*) in the ONH (RPC) improved significantly after PVI (before: 51.82 (48.41; 54.03); after: 52.49 (50.34; 55.62); *p* = 0.007). Flow density data in the different regions before and after PVI are summarized in Table 3.

After the evaluation of the funduscopic and angiographic data, one patient showed a silent microinfarction without visual impairment or visual field loss (Figure 1).

## 4. Discussion

The present study is the first of its kind to investigate ocular perfusion in AF patients before and after PVI. First, patients with AF showed a reduced FD compared to healthy controls, and rhythm control after PVI was associated with an improvement of retinal perfusion in individual patients as measured using OCT-A.

OCT-A is a novel technology, which enables the visualization of retinal microvasculature without intravenously injected dye. It is a non-contact technology and can be performed in an easy and fast manner. This technology has attracted a great deal of ophthalmological clinical research interest over the last two years and is finding increasing use in clinical practice [9,17,18,19]. Moreover, OCT-A also enables a quantitative analysis of blood flow in the retina and ONH. The repeatability and reproducibility of the quantitative analysis of flow density has been evaluated in detail before [10], and different studies in the literature have evaluated changes in different quantitative OCT-A parameters in different systemic diseases [20,21,22,23].

Since the retina is an embryological projection of the forebrain, and the ophthalmic artery arises from the carotid artery, OCT-A offers the opportunity to assess cerebral blood flow. In fact, various clinical studies have demonstrated a reduced cerebral perfusion in patients with AF. In AF patients, cardiac output decreases, thereby causing a reduction in the organs’ blood supply [7,8]. In a previous study by our group, we demonstrated a reduced ocular perfusion in patients with AF [13]. In this previous study, patients with AF at the time of imaging had significantly lower flow density in the RPC layer compared to patients with paroxysmal atrial fibrillation who had sinus rhythm at the time of imaging. However, both patient groups with atrial fibrillation showed a significantly lower flow density compared to healthy controls [13]. In fact, AF has been shown to be associated with a reduced cerebral blood flow velocity in the middle cerebral artery and with cognitive deficits in heart failure patients [6,7].

In patients with paroxysmal and persistent AF, catheter ablation with the goal of complete pulmonary vein isolation has been established as the cornerstone of rhythm control therapy [2]. However, the impact of PVI on ocular or cerebral perfusion has not yet been evaluated. In the present study, we demonstrated for the first time that ocular perfusion improved following PVI. The improved flow density was mainly observed in the RPC layer in the ONH, and improvement in the macula did not reach the level of statistical significance.

On the other hand, AF ablation itself is associated with a substantial risk for cerebral embolism and stroke [24,25,26]. In fact, several cerebral imaging studies have shown that AF ablation is associated with new silent cerebral lesions, the long-term effect of which is still unclear. None of the patients in our study developed any neurological or ocular symptoms. Interestingly, in one patient, a silent retinal artery occlusion was observed (Figure 1). Although the micro embolism was in the macular region in this particular case, the patient did not show any clinical symptoms. Although silent cerebral infarctions have been reported after PVI, to our knowledge this is the first case of retinal silent microinfarction after PVI reported so far.

In this context, it is also important to mention the association between AF and cognitive impairment. This association has been explained by silent microinfarction and cerebral hypoperfusion. In fact, mounting evidence suggests a correlation between brain perfusion and cognitive impairment [27,28,29]. In a previous study, we demonstrated a reduced FD in patients with Alzheimer’s disease and found a correlation between FD and the Fazekas scale in patients with Alzheimer’s disease [22]. The Fazekas scale is a well-established value that quantifies white matter lesions. It has a significant impact on the diagnosis of a vascular component during the diagnostic workup [22].

Indeed, small scale observational studies suggest a possible cognitive status improvement following pulmonary vein isolation [30]. However, this important issue is the subject of a controversial discussion and will have to be confirmed in further studies.

There are some important limitations to this study. Firstly, the present study was a non-randomized observational pilot study. Secondly, the study is limited by the small sample size. Third, due to the differences between the control group and the study group with regard to age and comorbidities, the comparison between the control group and the study group should be interpreted with caution. In fact, further studies with a longer follow up are needed to validate these findings.

## 5. Conclusions

In conclusion, this is the first study that demonstrates an improvement in ocular perfusion after rhythm control in patients with AF. Optical coherence tomography might be used to evaluate or monitor therapy success in patients with AF after rhythm control therapy with pulmonary vein isolation in patients with atrial fibrillation.

## Figures and Tables

**Figure 1 diagnostics-12-00038-f001:**
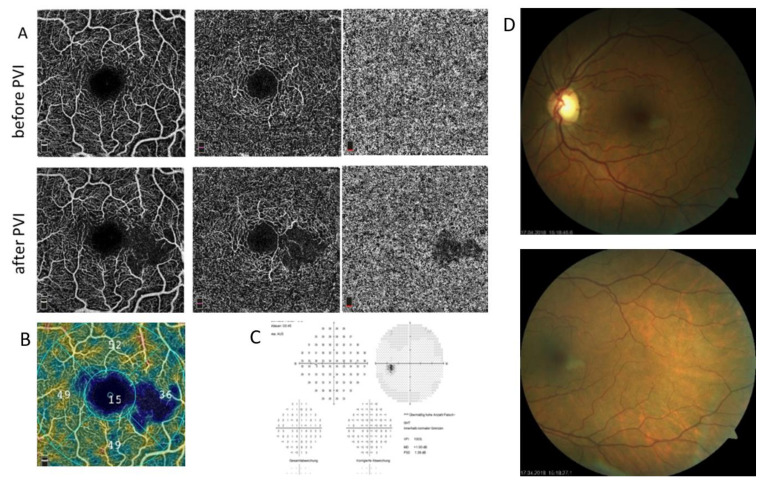
(**A**–**D**): Silent retinal microinfarction in a patient after PVI. (**A**): superficial and deep OCT angiogram of the macula before and after PVI. (**B**): Color-coded OCT angiogram of the macula showing the reduced FD temporal of the foveal avascular zone. (**C**): Visual field after PVI without significant defects. (**D**): Fundus photography images showing the small retinal edema. OCT = optical coherence tomography, PVI = pulmonary vein isolation, FD = flow density.

**Table 1 diagnostics-12-00038-t001:** Characteristics of the study population. AF = atrial fibrillation. IOP = intraocular pressure.

	Study Group	Control Group	*p*-Value
	Median (25th; 75th percentile)	Median (25th; 75th percentile)	
** *n* **	34	35	
**age (years)**	60.50 (52.00; 65.25)	50.70 (51.00; 70.00)	0.908
**sex (male/female)**	24/10	17/18	0.06
**spherical equivalent (diopters)**	0.375 (−1.31; 1.00)	0.500 (0.00; 1.50)	0.076
**IOP**	15.00 (14.00; 17.00)	15.00 (13.00; 17.00)	0.506
**visual acuity (decimals)**	1.00 (0.80; 1.00)	1.00 (0.80; 1.00)	0.651
**comorbidity**			
diabetes	2	0	
arterial hypertension	24	0	
hyperlipoproteinemia	7	0	

**Table 2 diagnostics-12-00038-t002:** Values of flow density (%) obtained in the regions indicated. Bold: statistically significant differences between AF patients and the control group.

	Study Group before Therapy (*n* = 34)	Control Group (*n* = 35)	*p*-Value
	Median (25th; 75th percentile)	Median (25th; 75th percentile)	
**OCT-A superficial**			
whole en face	48.77 (45.19; 52.12)	53.01 (50.00; 54.25)	**0.000**
fovea	26.65 (21.39; 30.95)	31.99 (26.94; 34.78)	**0.005**
parafovea	50.97 (47.09; 54.52)	54.78 (52.01; 56.34)	**0.001**
**OCT-A deep**			
whole en face	55.61 (49.40; 57.79)	57.15 (56.20; 58.64)	**0.005**
fovea	32.63 (28.89; 36.21)	29.45 (25.55; 32.18)	0.099
parafovea	57.22 (51.05; 60.43)	59.60 (58.12; 61.40)	**0.004**
**OCT-A RPC**			
whole en face	51.82 (48.41; 54.03)	56.00 (54.35; 57.70)	**0.000**
inside disc	47.83 (36.56; 52.47)	45.42 (38.79; 50.00)	0.670
peripapillary	57.02 (51.60; 63.35)	64.80 (61.74; 65.78)	**0.000**

**Table 3 diagnostics-12-00038-t003:** Values of flow density (%) before and after PVI obtained in the regions indicated. Bold: statistically significant differences. PVI = pulmonary vein isolation.

	before PVI	after PVI	*p*-Value
***n* = 34**	Median (25th; 75th percentile)	Median (25th; 75th percentile)	
**OCT-A superficial**			
whole en face	48.77 (45.19; 52.12)	50.30 (44.35; 52.73)	0.266
fovea	26.65 (21.39; 30.95)	29.28 (21.39; 31.90)	0.969
parafovea	50.97 (47.09; 54.52)	52.81 (46.80; 55.13)	0.221
**OCT-A deep**			
whole en face	55.61 (49.40; 57.79)	56.22 (48.70; 58.15)	0.166
fovea	32.63 (28.89; 36.21)	32.70 (27.19; 36.62)	0.881
parafovea	57.22 (51.05; 60.43)	58.21 (50.60; 61.02)	0.127
**OCT-A RPC**			
whole en face	51.82 (48.41; 54.03)	52.49 (50.34; 55.62)	**0.007**
inside Disc	47.83 (36.56; 52.47)	49.36 (36.24; 54.90)	**0.004**
peripapillary	57.02 (51.60; 63.35)	60.98 (54.03; 65.89)	**0.008**

## Data Availability

The data presented in this study are available on request from the corresponding author.

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
