# Peer review of "Improvement of Retinal Microcirculation after Pulmonary Vein Isolation in Patients with Atrial Fibrillation—An Optical Coherence Tomography Angiography Study"

_diagnostics, 2021, doi:10.3390/diagnostics12010038_

Round 1

Reviewer 1 Report

This is a well-written article on OCT-A in patients with atrial fibrillation (AF).

The findings of the authors that patients with AF have a decreased flow density in the retina and of the optic nerve head compared to controls are very interesting. Moreover, flow density seems to improve after treatment of AFIB. 
Unfortunately, no follow-up of the control group has been reported. The difference although statistically significant are small and thus it may be interesting to know whether fluctuations are also observable in the control group. If these data of the control group exist, they should be included in the manuscript.

Furthermore, the control group has more comorbodities than the study group such as diabetes. This may have skewed the results.

The male/female ratio of the control and the study group should be reported as well.

Finally, the control group seems to be you younger than the study group. The median age of the control group ist 10 years lower than the study group. Albeit the fact that there seems no statistical difference with respect to age of both groups this may also be due to the rather small number of patients in each group.

Author Response

Thank you for your constructive comment. In this study, we have not performed a follow-up of the control group. The study group (n=34 patients) consisted of 24 men and 10 woman. The control group (n=35 patients) included 17 men and 18 women (p>0.05, χ square test).

In fact, the control group has more comorbidities than the study group, and the median age of the control group is lower than the median age of the study group. For that reason, any comparison of flow densities between the control group and the study group should be interpreted with caution. In order to study the impact of atrial fibrillation on retinal perfusion without confounders, we performed a follow up OCT-A within the study group. These results support the notion that atrial fibrillation has indeed an influence on retinal perfusion measured by OCT-A.

Reviewer 2 Report

The article is very interesting. Authors describe a new use of ophthalmic diagnostic devices for evaluating patients with internal diseases. It can contribute to greater cooperation of ophthalmologists with other specialists. I recommend the article for publication. I would appreciate it if the authors can add information on how many patients were excluded from the study due to media opacities, VR pathologies or neurological diseases (line 74-75).  Please also add information about a number of patients excluded from the quantitative analysis due to poor quality images (line 119)? Do you think those excluded data could influence your results?

Author Response

Thank you very much for your instructive and important comment. In fact, patients with media opacities, VR pathologies or neurological diseases were not considered to become study participants. This issue has been clarified in the methods section. One patient had to be excluded from the study due to a poor imaging quality. In the authors opinion, this has had no relevant influence on the study results.